# Impact of Neoadjuvant Treatment on Target Expression in Rectal Cancer for Near-Infrared Tumor Imaging

**DOI:** 10.3390/cancers17121958

**Published:** 2025-06-12

**Authors:** Elham Zonoobi, Lisanne K. A. Neijenhuis, Annelieke A. Lemij, Daan G. J. Linders, Ehsan Nazemalhosseini-Mojarad, Shadhvi S. Bhairosingh, N. Geeske Dekker-Ensink, Ronald L. P. van Vlierberghe, Koen C. M. J. Peeters, Fabian A. Holman, Rob A. E. M. Tollenaar, Denise E. Hilling, A. Stijn L. P. Crobach, Alexander L. Vahrmeijer, Peter J. K. Kuppen

**Affiliations:** 1Department of Surgery, Leiden University Medical Center, P.O. Box 9600, 2300 RC Leiden, The Netherlands; e.zonoobi@lumc.nl (E.Z.); l.k.a.neijenhuis@lumc.nl (L.K.A.N.); a.a.lemij@lumc.nl (A.A.L.); d.g.j.linders@lumc.nl (D.G.J.L.); e.nazemalhosseini_mojarad@lumc.nl (E.N.-M.); s.bhairosingh@lumc.nl (S.S.B.); n.g.dekker-ensink@lumc.nl (N.G.D.-E.); r.l.p.van_vlierberghe@lumc.nl (R.L.P.v.V.); k.c.m.j.peeters@lumc.nl (K.C.M.J.P.); f.a.holman@lumc.nl (F.A.H.); r.a.e.m.tollenaar@lumc.nl (R.A.E.M.T.); d.hilling@erasmusmc.nl (D.E.H.); p.j.k.kuppen@lumc.nl (P.J.K.K.); 2Division of Gastroenterology and Hepatology, Department of Medicine, School of Medicine, Stanford University, Stanford, CA 94305, USA; 3Centre for Human Drug Research, 2333 CL Leiden, The Netherlands; 4Department of Surgical Oncology and Gastrointestinal Surgery, Erasmus MC Cancer Institute, University Medical Center Rotterdam, 3015 GD Rotterdam, The Netherlands; 5Department of Pathology, Leiden University Medical Center, 2333 ZA Leiden, The Netherlands; a.s.l.p.crobach@lumc.nl

**Keywords:** rectal cancer, molecular imaging, near-infrared fluorescence imaging, target selection, tumor imaging, neoadjuvant therapy

## Abstract

Rectal cancer is a common and clinically challenging disease that typically requires a combination of preoperative chemoradiotherapy and surgery. In patients who respond well to preoperative treatment, a non-operative “watch-and-wait” approach may be considered to avoid the complexity associated with surgery. However, accurately identifying patients with complete tumor response remains difficult due to limitations in tumor-imaging techniques. This study investigated the potential of near-infrared fluorescence imaging to improve detection of residual tumor tissue during endoscopy. We analyzed paired tissue samples from 51 rectal cancer patients taken before and after chemoradiotherapy to evaluate the expression of specific molecular markers. Our findings revealed that CEA and c-MET remained consistently overexpressed in tumor tissue and were minimally affected by treatment. These results support their potential as reliable imaging targets. Further clinical studies are warranted to validate these markers and advance fluorescent tumor imaging strategies for improved treatment planning and patient selection.

## 1. Introduction

Rectal cancer (RC) presents a significant treatment challenge, as efforts continue to reduce the risk of distant metastases, preserve quality of life, and personalize treatment by identifying patients who respond to neoadjuvant therapy [1]. The standard approach for locally advanced RC (LARC, stage II-III) involves neoadjuvant chemoradiotherapy (nCRT) followed by total mesorectal excision (TME). Although this protocol effectively controls tumor progression, it is often associated with significant morbidities from postoperative complications [1,2]. In recent years, clinical trials such as RAPIDO [3] and PRODIGE−23 [4] have highlighted the benefits of total neoadjuvant treatment (TNT), demonstrating enhanced control over distant metastases and achieving a pathological complete response (pCR) for the primary tumor in up to 30% of patients. Consequently, TNT is gaining widespread acceptance as a cornerstone in the treatment of locally advanced cases, particularly given the substantial proportion of patients who achieve pCR following this approach [5]. The variability in tumor response has sparked ongoing discussions about the potential for more individualized approaches, including organ-preserving strategies for selected patients [5].

Selecting patients for watchful waiting or organ preservation requires a highly accurate evaluation of their clinical response to nCRT to ensure optimal treatment outcomes. Current imaging modalities, including magnetic resonance imaging (MRI), white-light colonoscopy, and digital rectal examination, face significant limitations in distinguishing between residual microscopic cancer cells and extensive post-treatment fibrosis [1,6,7,8]. These challenges lead to overtreatment, where patients with near-complete responses may be denied organ preservation options and will therefore undergo total mesorectal excision (TME) and have a pathological complete response (pCR) at final pathology. In trials like RAPIDO and PRODIGE−23, 27–28% of patients who underwent TME after neoadjuvant therapy were found to have pCR. Conversely, inadequate response assessment also contributes to significant rates of local regrowth in watch-and-wait (W&W) patients, with up to one-third ultimately requiring TME due to tumor recurrence [9,10]. These issues highlight an urgent need for advanced imaging technologies capable of precisely identifying viable cancer cells while avoiding misclassification of normal or fibrotic tissue. Such advancements could revolutionize decision-making in W&W strategies, ensuring that patients with near-complete clinical responses (near-cCR) receive tailored and appropriate management.

Tumor-specific fluorescence imaging facilitates real-time visualization of tumor cells within the near-infrared (NIR) spectrum (700–900 nm). This method employs a tracer comprising a fluorophore linked to a targeting agent, which either binds to tumor-specific ligands [11] or responds to the tumor microenvironment [12]. Excitation by a tailored light source prompts the fluorophore to emit photons, captured by a specialized camera, enabling precise optical imaging of cells expressing distinct molecular markers [11,13].

For a tumor-targeted fluorescence tracer to effectively delineate cancerous tissue, it is essential that the targeted biomarker is significantly overexpressed in tumor cells while being absent or minimally expressed in healthy surrounding tissue and fibrotic regions. Prior studies have pinpointed several tumor-associated targets including carcinoembryonic antigen-related cell adhesion molecule 5 (CEACAM5, referred to as CEA) [14,15], epithelial cell adhesion molecule (EpCAM) [16,17], mesenchymal–epithelial transition factor (c-MET) [18,19,20], and epidermal growth factor receptor (EGFR) [21,22], which are overexpressed in most colorectal tumors and serve as key biomarkers in colorectal cancer (CRC). As such, these well-established biomarkers have clinically available NIR fluorescence tracers [18,23,24,25]. Similar work in esophageal cancer by Galema et al. has demonstrated that neoadjuvant therapy can markedly influence target expression for NIR fluorescence imaging, reinforcing the need to assess these changes within each cancer type [26].

This study aims to identify clinically relevant and reliable molecular markers for tumor-specific NIR fluorescence imaging to improve the detection of residual disease and enhance the precision of endoscopic assessment in future RC patients. Specifically, the study evaluates the effect of nCRT on the expression levels of these markers by comparing biopsies obtained before treatment and corresponding primary tumor specimens from resection samples in different treatment response groups as determined by using the Mandard (TRG) classification [5,27]. These findings may support the integration of targeted imaging in selecting patients eligible for the W&W approach.

## 2. Materials and Methods

### 2.1. Patient Cohort

This study was approved by the Medical Ethical Committee of the Leiden University Medical Center (METC LDD LUMC, code# B20.052 approved on 17 December 2020) and was conducted according to the Declaration of Helsinki. Formalin-fixed paraffin-embedded (FFPE) tissue blocks from RC patients who underwent surgical treatment were retrieved and evaluated by a specialized pathologist (A.S.L.P.C.). Tissue blocks containing both tumor and adjacent healthy tissue were selected for analysis when available; in total, 19 such blocks from a larger cohort of 59 patients were included, allowing for direct comparative evaluation. These 19 patients were selected based on the availability of paired diagnostic biopsy and resection specimens containing both tumor and adjacent normal mucosa. Selection was based on tissue quality and completeness of clinical annotation, rather than randomization. Diagnostic biopsy samples obtained before neoadjuvant therapy were paired with tissue blocks from corresponding resection specimens collected post-treatment. Tissue slides were chosen from patients exhibiting a pathological complete response (Mandard tumor regression grade (TRG) 1), a near-complete response (Mandard TRG 2), a moderate response (Mandard TRG 3), or minimal response (Mandard TRG 4–5) [5,27]. For cases with a pathological complete response, where no residual tumor cells were detected in the resection specimens, tissue slides were selected based on reactive changes such as fibrosis. All tissue blocks were obtained from the tissue bank at Leiden University Medical Center (LUMC) and included samples from patients who underwent surgery between 2007 and 2022. Only cases with adequate paired tumor and adjacent normal mucosa were included for tumor-to-normal (T/N) ratio analyses, which limited the number of evaluable samples for this component of the study. The time interval between neoadjuvant therapy and surgery ranged from 1 to 91 weeks and was recorded for all patients; however, due to sample size constraints, this variable was not stratified in the analysis.

### 2.2. Immunohistochemistry

For each patient, several consecutive 4 µm sections were prepared from the FFPE block and affixed to adhesive slides (Superfrost Plus adhesion microscope slides, Epredia, Portsmouth, NH, USA). These sections underwent immunohistochemical staining to determine the expression level of the set of 4 chosen markers, alongside H&E staining for histological assessment. The slides were deparaffinized using xylene and subsequently rehydrated through a series of ethanol solutions with decreasing concentrations. Following a rinse with demineralized water, the slides were treated with a solution of 0.3% hydrogen peroxidase (Merck Millipore, Darmstadt, Germany) for 20 min at room temperature to neutralize endogenous peroxidase activity. Antigen retrieval was conducted using techniques tailored to each type of primary antibodies (Table 1). After antigen retrieval, the slides were washed with phosphate-buffered saline (PBS, pH 7.5) and incubated overnight at room temperature with the primary antibodies, which had been diluted to optimal concentrations based on previous testing with positive controls and CRC tissues. Post-incubation, the slides were washed with PBS and then treated with HRP-labeled corresponding secondary antibodies (EnVision, Agilent Technologies, Glostrup, Denmark) for 30 min at room temperature (Table 1). The immunostaining was visualized using a 10 min incubation with 3,3′-diaminobenzidine (DAB, Agilent, Santa Clara, CA, USA) at room temperature, followed by a 20 s counterstain with hematoxylin (VWR International, Amsterdam, The Netherlands). The slides were then dehydrated at 37 °C and sealed with a coverslip using pertex mounting medium. Finally, the stained slides were digitized using the 3D-Histech 250 midi scanner (3D-Histech Ltd., Budapest, Hungary).

### 2.3. Scoring Method

Biopsies and primary resection samples from RC patients who underwent neoadjuvant therapy were evaluated for the expression of CEA, EpCAM, c-MET, and EGFR. The biomarker expression was evaluated by Dr. A.S.L.P. Crobach, a certified gastrointestinal pathologist, using the histological score (H-score) method, without prior knowledge of the patient’s details. Immunohistochemically stained tissue sections were rated based on four levels of staining intensity: none, weak, moderate, and strong. The H-score was calculated by multiplying the percentage of the positive area by its respective intensity (1 × % positive + 2 × % positive + 3 × % positive) and dividing the sum by 100. This method quantifies overall expression levels, with scores ranging from 0 (no expression) to 3 (100% strong expression) [28]. Tumor tissue expression was evaluated across all samples, while adjacent normal epithelial tissue was assessed in five representative samples from each TRG group. Representative samples were selected to include both healthy and tumor tissues from diagnostic biopsies (pre-treatment samples) and corresponding resection specimens (post-treatment samples). A T/N ratio of the H-score was calculated for each slide where both tumor and adjacent healthy mucosa were assessed, with a ratio of ≥2 considered indicative of adequate contrast. Due to limited availability of adjacent normal mucosa in all samples, T/N ratio assessments were only possible in a subset of cases that met these tissue requirements.

### 2.4. Statistical Analysis

Statistical analyses were conducted using SPSS version 29.0 (SPSS©; IBM Corporation, NY, USA) and GraphPad Prism 6 (GraphPad Software Inc., La Jolla, CA, USA). Descriptive statistics were used to summarize patient characteristics. Differences in marker expression between tumor tissues and adjacent normal epithelial tissues were assessed using the Friedman test, both pre-and post-treatment. Changes in biomarker expression within tumor tissues, comparing biopsy and resection specimens, as well as the effects of nCRT on H-scores, were evaluated using the Wilcoxon signed-rank test for paired tissue samples obtained from the same patients. To explore variations in marker expression across Tumor Regression Grade (TRG) cohorts (TRG1, TRG2 vs. TRG3 vs. TRG4/5), the Wilcoxon signed-rank test was again applied for paired samples. Statistical significance was set at *p* ≤ 0.05 for all analyses.

## 3. Results

Tissue samples from 59 patients were retrospectively analyzed. Archived pre-treatment biopsies (*n* = 59), originally collected prior to the initiation of nCRT, were retrieved alongside corresponding post-treatment surgical specimens (*n* = 59) obtained after completion of nCRT. In total, 8 of the 59 paired samples (biopsies and resections) were excluded from further analysis due to inadequate tissue quality or lack of tumor tissue. Patient and tumor characteristics of included patients are summarized in Table 2. Patients received different forms of neoadjuvant therapy, either as part of a long-course or short-course strategy. Long-course chemoradiotherapy (LCCRT) was administered in 26 patients, consisting of capecitabine given concurrently with 25 × 2 Gy fractions of radiotherapy. Short-course radiotherapy (SCRT), defined as 5 × 5 Gy, was administered in 16 patients. In all of these cases, chemotherapy (CapOx or CapOx plus bevacizumab) was given sequentially either before or after RT as part of a TNT regimen. No patients received chemotherapy concurrently with SCRT. Additional neoadjuvant regimens included five patients treated with CapOx/Avastin and SCRT, two with CapOx and SCRT, one with a 13 × 3 Gy radiotherapy protocol, and one with CapOx/Avastin followed by other agents. Overall, 33 of 51 patients received neoadjuvant chemotherapy in combination with radiotherapy consistent with TNT. Details of neoadjuvant regimens are shown in Table 2. The median interval between neoadjuvant therapy and surgery was 15.1 weeks (range: 1.0–91.1 weeks; interquartile range: 13.7 weeks). The majority of tumors were adenocarcinomas.

Tumor regression grades (TRGs) were assessed using the Mandard classification system and categorized into TRG1, TRG2, TRG3, and TRG4/5 groups. Among the 51 evaluable patients, 15 achieved a pathological complete response (pCR, TRG1). TRG2 (*n* = 8) and TRG3 (*n* = 13) represented near-complete and moderate tumor regression, respectively. Fifteen tumors showed no response to nCRT (TRG4/5).

### 3.1. Immunohistochemical Staining Pattern of CEA, EpCAM, c-MET, and EGFR Expression in Rectal Cancer Biopsies

Staining patterns for all selected markers, including CEA, EpCAM, c-MET and EGFR, were predominantly heterogeneous in both tumor and adjacent normal epithelium, exhibiting intra- and inter-sample variations, whereas c-MET staining was mostly homogeneous (intra-sample variation) in normal tissues (Figure 1). CEA, EpCAM, and EGFR were consistently expressed in normal epithelial cells across nearly all RC samples, with varying expression levels. CEA and c-MET were predominantly localized to the apical and luminal surfaces, especially in mucin-producing cells, with expression diminishing in deeper layers (Table 3). EpCAM exhibited moderate to strong membranous staining in over 90% of normal epithelial cells, reflecting its stable presence in the epithelial lining. In contrast, EGFR showed weak but uniform cytoplasmic expression throughout the muscular layers, extending into deeper tissues (Table 3). Notably, in pCR cases, EGFR expression was also detected in fibrotic tissues.

### 3.2. CEA, EpCAM, c-MET, and EGFR Expression Before Neoadjuvant Therapy

Sequential tissue sections from biopsy samples were stained for CEA, EpCAM, c-MET, and EGFR, with tumor expression levels quantified using the H-score method. Figure 2 presents representative IHC staining images of these four targets from an RC patient, comparing biomarker expression in pre-nCRT biopsy and post-nCRT resection tissues. CEA tumor expression was observed in all biopsy samples (51/51, 100%), with staining intensities ranging from weak to strong. Similarly, EpCAM was expressed in 98.4% of the biopsy samples (50/51), showing comparable variability in expression levels. c-MET staining was positive in 92.1% of the biopsy samples (47/51), with weak to moderate expression, while 7.8% of the cases (4/51) displayed no c-MET expression. EGFR expression was detected in 90.2% of the biopsy samples (46/51), predominantly at weak levels, with 9.8% of the cases (5/51) showing no detectable expression.

To assess the potential of molecular fluorescence imaging before nCRT, biopsy samples from both tumors and adjacent healthy mucosa were analyzed to determine the T/N H-score ratio (Figure 3A). A ratio of ≥2 was considered indicative of sufficient contrast. Among the biomarkers assessed, c-MET exhibited the highest median T/N expression ratio, with 52.6% of samples exceeding the threshold of 2. In contrast, CEA and EpCAM had median T/N ratios of 1.53 and 1.36, respectively, with only 26.3% and 15.8% of samples showing ratios above 2. EGFR was not overexpressed in RC biopsies compared to adjacent healthy epithelium, with none of the samples exhibiting a T/N ratio greater than 2. The expression scores for all biomarkers across the tumor biopsies are summarized in Table 4.

### 3.3. Effect of Neoadjuvant Chemoradiotherapy on Marker Expression in Adjacent Healthy Mucosa

To identify optimal fluorescence-guided imaging targets, assessing nCRT-induced changes in biomarker expression within adjacent healthy mucosa is essential. Accordingly, we quantified CEA, EpCAM, c-MET, and EGFR levels in pre-treatment biopsies and post-treatment resection specimens (Table 4), with results from a representative patient shown in Figure 2. Following nCRT, CEA, EpCAM, and EGFR expression increased in normal mucosa, while c-MET remained low. CEA showed a shift toward moderate expression post-treatment (44.4% vs. 26.3% pre-treatment), while EpCAM displayed a higher proportion of moderate-to-strong expression (77.8% post-treatment vs. 68.4% pre-treatment). Similarly, EGFR increased in moderate expression (27.8% post-treatment vs. 5.3% pre-treatment). In contrast, c-MET remained minimally expressed, with no or weak expression in all cases. These findings indicate that nCRT enhances CEA, EpCAM, and EGFR expression in normal mucosa, which may impact their specificity as imaging targets, whereas c-MET expression remains largely unchanged, suggesting a more stable marker post-treatment (Figure 2).

### 3.4. Comparative Expression Patterns of Biomarkers in Tumor and Normal Tissues Before and After nCRT

H-scores of CEA, EpCAM, c-MET, and EGFR were compared between tumor and adjacent normal epithelial tissues using pre- and post-treatment specimens (Figure 4A,B). Following nCRT, CEA expression in tumor tissues increased significantly in comparison to normal tissues (*p* = 0.003), with T/N ratios ≥ 2 observed in 45.5% of cases (Figure 3B), indicating its potential as a fluorescence-guided imaging target. c-MET expression in tumors also exhibited a significant increase (*p* = 0.009), with ratios ≥ 2 in 45.5% of cases, supporting its utility in imaging applications. EpCAM expression rose in both tumor and normal tissues (*p* = 0.045), though only 9.1% of cases demonstrated a T/N ratio ≥ 2. In contrast, EGFR expression remained elevated in normal tissues, with a non-significant increase in tumors (*p* = 0.929), suggesting limited tumor-specific imaging potential. To enhance tumor detection coverage, a complementary combination of CEA and c-MET was considered. As shown in Figure 3C, 36.4% (4/11) of tumors demonstrated high contrast (ratio ≥ 2) for both markers, while 18.2% (2/11) were detectable by only one. However, 45.5% (5/11) of tumors showed limited detectability, with T/N ratios < 2 for both markers. Notably, CEA, EpCAM, and EGFR expression were observed in normal epithelial cells in cases achieving pCR. These results demonstrate that nCRT enhances CEA and c-MET expression in tumor tissues, while EpCAM shows moderate differentiation, and EGFR remains predominantly expressed in normal epithelium, thus restricting its tumor-specific utility post-treatment.

### 3.5. Effects of Neoadjuvant Chemoradiotherapy on Tumor Marker Expression Among the Different Treatment Response Groups

In patients achieving a pCR (TRG1), no residual tumor was detectable in surgical specimens, precluding marker expression analysis. Among near-complete responders (TRG2), changes in marker expression were minimal and statistically non-significant (Figure 5). In moderate responders (TRG3), significant increases were observed for CEA (from 2.2 to 2.7, *p* = 0.049) and EpCAM (from 1.8 to 2.4, *p* < 0.05), whereas EGFR expression remained unchanged (*p* > 0.05). For tumors with minimal or no response (TRG4/5), significant upregulation was noted across all markers post-nCRT: CEA (*p* = 0.004), EpCAM (*p* = 0.002), c-MET (*p* = 0.003), and EGFR (*p* = 0.001). These data demonstrate that marker expression varies with tumor response to nCRT, with the most pronounced increases occurring in poorly responsive tumors (TRG4/5). Figure 5 visually summarizes these differential expression patterns across TRG categories.

## 4. Discussion

In this study, the usability of CEA, EpCAM, c-MET, and EGFR as targets for fluorescence imaging in RC after neoadjuvant treatment was evaluated through IHC. Among the markers assessed, CEA, EpCAM, and c-MET demonstrated significant overexpression in rectal tumors. While more than 90% of the tumor specimens showed positive staining for these markers, only a subset exhibited a T/N expression ratio of ≥2, which is considered the threshold for adequate imaging contrast. This discrepancy highlights that although overall overexpression is prevalent, only a proportion of cases exhibit sufficient differential contrast for effective fluorescence-guided surgery. The number of patients who showed T/N expression ratios above 2 remained the same. Based on the percentage of samples with a T/N expression ratio higher than 2, c-MET and CEA exhibited the greatest differential contrast after nCRT, highlighting their potential as targets for molecular tumor-guided imaging in RC. These findings were derived from cases in which paired tumor and adjacent normal mucosa were available; therefore, T/N ratio assessments were only possible in a subset of the total patient cohort. This limitation may influence the generalizability of the ratio-based conclusions.

Response evaluation following neoadjuvant therapy using targeted NIR fluorescence endoscopy holds considerable potential for improving the precise identification of RC patients who may benefit from the W&W strategy. This approach has the potential to prevent unnecessary surgeries and minimize the risk of local tumor regrowth. Our finding of abundant expression of CEA, c-MET, and EpCAM in most rectal tumors corroborates previous large-scale studies [29,30,31,32]. While EpCAM is frequently overexpressed in tumors, its relatively low T/N ratio limits its utility as a marker for distinguishing microscopic residual disease, consistent with earlier observations [32]. In our study, resection specimens with a pCR, CEA and EpCAM exhibited moderate expression on the surface of normal epithelial tissue, while c-MET demonstrated minimal to no expression in normal tissues. Similarly, Boogerd et al. reported mild, uniform positivity for CEA and EpCAM in the tumor bed of pCR specimens, with c-MET expression confined to (healthy epithelial) mucin-producing cells and nonspecific staining in the muscularis propria [32]. These patterns may reflect the distinct biological roles of these markers, with CEA and EpCAM integral to cell adhesion and epithelial integrity, accounting for their residual expression in normal tissues and the tumor bed post-nCRT [33].

In TRG2 cases, EpCAM expression showed no significant change in post-treatment specimens compared to pre-treatment biopsies. Similarly, CEA and c-MET expression levels showed no significant variation in TRG2 responders, a finding that may be influenced by the limited sample size in this subgroup. EpCAM overexpression has been linked to poor prognostic features, including larger tumor size, lymph node metastasis, and poor outcomes, as reported in a meta-analysis of gastric cancer studies by Dai et al. [34]. In our study, EpCAM showed a limited T/N expression ratio (<2), suggesting insufficient differential contrast for effective fluorescence imaging of microscopic residual disease. Additionally, our data demonstrated that CEA and EpCAM remained overexpressed in tumors from TRG3 and TRG4/5 post nCRT, aligning with previous findings reporting high expression levels in 93% and 100% of partial- and nonresponders, respectively [32]. Furthermore, we observed a notable overexpression of c-MET in tumors from nonresponders following nCRT, highlighting its potential relevance as a molecular imaging target. EGFR, a transmembrane protein linked to CRC [35], showed low tumor expression but higher levels in normal mucosa, scar tissue, and muscle post-nCRT in our study, reducing its specificity as an FGS target. This variability may also be partly influenced by the wide range in time between biopsy and resection (1–91 weeks), which was not stratified in the analysis and could have affected marker expression dynamics. This suggests nCRT may shift EGFR distribution, hindering tumor visualization. Its variability and inconsistent prognostic value [36,37] further limit its reliability for such applications. Tiernan et al. reported EGFR overexpression in only 32.8% of 280 colorectal tumor cases compared to normal tissue, far less consistent than CEA’s 98.8%, underscoring its limited specificity for in vivo targeting [22]. Similarly, a study with panitumumab-IRDye800CW found EGFR expression did not directly correlate with the tumor-to-background fluorescent signal ratio, due to factors like tumor vascularity and probe penetration affecting fluorescence beyond receptor levels [38]. DeLong et al. also position EGFR as a promising yet challenging target, noting its overexpression in CRC but emphasizing the need for high T/N contrast, which may be elusive given tissue variability [21]. Collectively, these findings underscore EGFR’s limitations for FGS, particularly post-nCRT, suggesting exploration of more specific targets.

The integration of near-infrared (NIR) fluorescence endoscopy with targeted contrast agents offers a promising approach to enhance response evaluation post-nCRT, enabling precise identification of complete responders and supporting the W&W strategy to reduce unnecessary surgeries. In this study, c-MET and CEA emerged as leading targets for fluorescence-guided imaging, building on their established overexpression and differential contrast [18,19,23]. This potential is being actively explored in our ongoing clinical trial (#NCT06280690), which assesses the feasibility of SGM−101, a CEA-targeted agent, for endoscopic detection of malignant rectal polyps. Preliminary results from this trial are anticipated to provide critical insights into its clinical applicability. Similarly, VB5–845D−800CW, an EpCAM-targeting tracer, demonstrated safety and feasibility in a recent first-in-human trial conducted by our group [24]. Although its role in RC NIR imaging remains untested, forthcoming trial outcomes may clarify its utility. Comparable translational work was recently reported by Galema et al., who investigated the impact of neoadjuvant therapy on target expression for NIR imaging in esophageal cancer. Their findings support the feasibility of imaging-based response monitoring across gastrointestinal tumors and further underscore the relevance of our study in the rectal cancer setting [26]. To date, no clinical studies have evaluated these markers for post-nCRT response assessment, positioning our findings as a pivotal foundation for future validation and optimization.

Despite the noteworthy results, this study has some limitations that should be acknowledged. The semiquantitative nature of IHC for measuring protein expression poses challenges. Although IHC is widely used and well-established, it is not fully quantitative and can be influenced by factors such as antibody selection, tissue fixation methods, and staining protocols, leading to variability in staining interpretation. While validated antibodies and a published scoring system [28] were employed, artifacts and microscopic tumors in TRG2 cases posed additional challenges, leading to the exclusion of up to eight patients’ samples. However, this minimal exclusion is unlikely to have significantly impacted the overall findings. Furthermore, the correlation between IHC-derived T/N ratios and clinical TBR is not always direct; a T/N ratio of ≥2 suggests potential for fluorescence-guided imaging, but clinical TBR can vary due to physiological and technical factors. The correlation between the expression of markers such as CEA, EpCAM, c-MET, and EGFR, as measured by IHC, and their in vivo fluorescence signal intensity using NIRF probes directed against these biomarkers remains unclear and requires further investigation. Notably, prior studies have demonstrated a strong correlation between in vivo fluorescence signal intensity and IHC-based expression of c-MET in untreated tumors [19,39], highlighting the translational potential of these imaging targets. Future research should investigate this correlation in the post-nCRT setting to optimize the clinical utility of these biomarkers for fluorescence-guided imaging in RC. Furthermore, clinical tracers may differ from IHC antibodies in terms of epitope targeting and affinity, necessitating careful evaluation during translation. To address these challenges, the next critical step involves conducting in vivo feasibility studies to validate the findings and refine imaging protocols. Our data suggest that CEA and c-MET, recognized as the most promising targets, may effectively distinguish tumor from surrounding normal tissue, in our study in about half of the evaluated specimens. This restricted performance highlights the necessity of exploring complementary targeting strategies or broad-spectrum agents to enhance detection accuracy and address the persistent challenge of tumor heterogeneity in RC. While standard histopathology provides definitive insight into tumor regression after resection, it cannot inform real-time clinical decision-making prior to or during surgery. In contrast, tumor-targeted fluorescence imaging has the potential to detect residual disease in vivo during endoscopic assessment, particularly in near-complete responders. Therefore, molecular markers such as CEA and c-MET may serve as crucial adjuncts to conventional imaging by guiding preoperative response evaluation and patient selection for organ-preserving strategies.

## 5. Conclusions

Our study demonstrated that CEA, EpCAM, and c-MET are abundantly expressed in rectal cancer tumors, with nCRT exerting minimal impact on their expression in both tumor and adjacent normal tissues. Among these markers, CEA and c-MET showed the greatest promise for NIR fluorescence imaging due to their sustained differential expression post-nCR. These characteristics highlight their potential as reliable imaging targets. Determining the optimal biomarker lays a foundation for clinical practice, and further investigation, including comparative clinical trials, is essential to validate their clinical applicability and efficacy. These findings provide a vital basis for developing tumor-specific NIR fluorescence imaging strategies aimed at enhancing the detection of residual RC after neoadjuvant therapy.

## Figures and Tables

**Figure 1 cancers-17-01958-f001:**
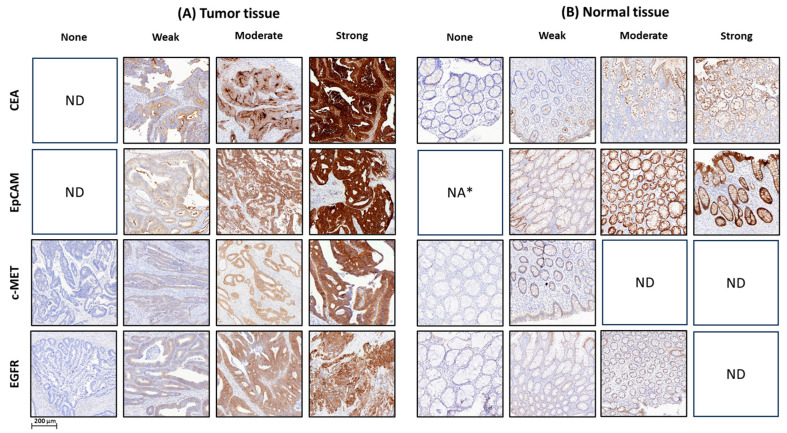
Immunohistochemical H-score classifications for CEA, EpCAM, c-MET, and EGFR in rectal cancer tissues. Representative images illustrate immunohistochemical (IHC) H-score classifications for CEA, EpCAM, c-MET, and EGFR, quantifying expression levels in tumor and adjacent normal epithelial tissues of rectal cancer patients. Panels depict H-scores ranging from none (no expression) to strong (strong expression), applied uniformly to both tissue types, with distinct staining patterns selected from different biopsy and resection samples. (**A**) Tumor tissues and (**B**) adjacent normal tissues illustrate varying expression profiles, exemplifying the scoring methodology. Abbreviations: CEA, carcinoembryonic antigen; EpCAM, epithelial cell adhesion molecule c-MET, mesenchymal–epithelial transition factor; EGFR, epidermal growth factor receptor; ND, not detected; NA*, not available due to a technical error.

**Figure 2 cancers-17-01958-f002:**
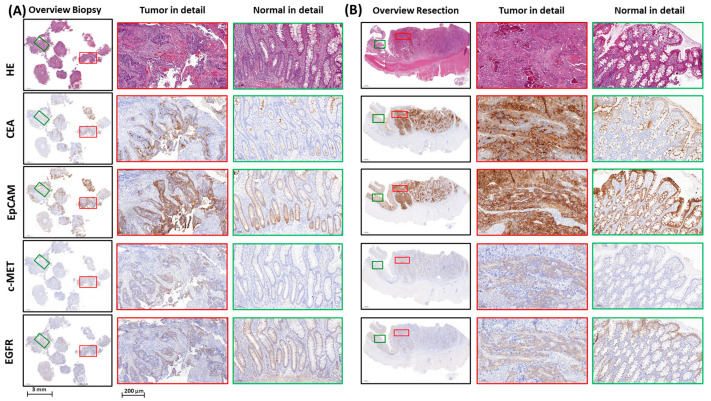
Representative images of immunohistochemical stainings of CEA, EpCAM, c-MET, and EGFR expression in biopsy and resection tissues of a rectal cancer patient. Representative images of tumor marker expression in diagnostic biopsy tissue before nCRT (**A**) and resection tissue after nCRT (**B**) of a rectal cancer patient are shown. Panels include sequential tissue slides stained for CEA, EpCAM, c-MET, EGFR, and corresponding HE staining. In the biopsy column (**A**), the first (leftmost) subpanel displays an overview of the biopsy sample, the second subpanel shows detailed tumor tissue, and the third subpanel illustrates detailed normal tissue. Similarly, in the resection column (**B**), the first (leftmost) subpanel provides an overview of the resected tissue, while the second and third subpanels present detailed tumor and normal tissue, respectively. Red rectangles mark tumor regions, and green rectangles mark adjacent normal mucosa used for higher-magnification panels. Scale bars represent 3 mm in overview images and 200 μm in detailed panels. Abbreviations: HE, hematoxylin and eosin; CEA, carcinoembryonic antigen; EpCAM, epithelial cell adhesion molecule; c-MET, mesenchymal– epithelial transition factor; EGFR, epidermal growth factor receptor; nCRT, neoadjuvant chemoradiotherapy.

**Figure 3 cancers-17-01958-f003:**
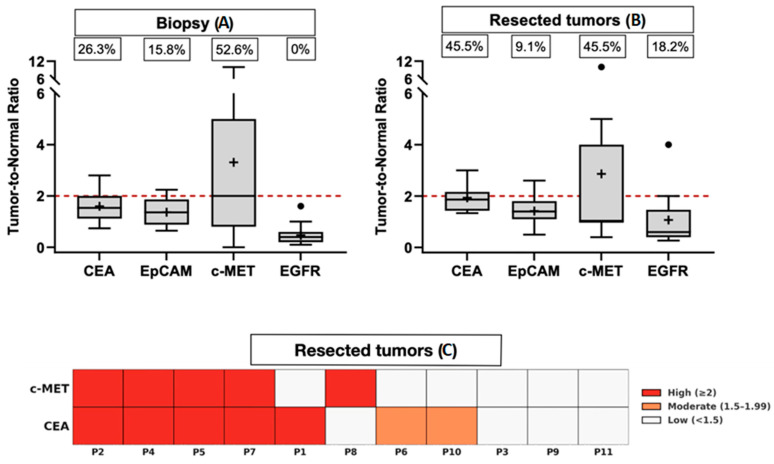
Tumor-to-normal tissue ratio of tumor markers in biopsy and resection tissues of rectal cancer. The boxplots illustrate the tumor-to-normal mucosa H-score ratio for each tumor marker in biopsy (**A**) and resection tissues (**B**). Each boxplot displays the minimum, first quartile (Q1), median, third quartile (Q3), and maximum values, with error bars representing the range. The mean value is denoted by a plus sign (+). The percentages above the boxplot indicate the proportion of tissue samples with a tumor-to-normal mucosa ratio ≥ 2.0. The red dashed line marks a tumor-to-normal mucosa ratio of 2, emphasizing the threshold for adequate contrast. (**C**) Heatmap of tumor-to-normal expression ratios for CEA and c-MET in 11 resection specimens following neoadjuvant treatment. Only patients with available paired tumor and adjacent normal tissue for both markers were included in this analysis (*n* = 11). Each column represents one patient (P1–11); rows indicate biomarker expression. Color intensity reflects the tumor-to-normal ratio: red indicates high expression (≥2), orange moderate expression (1.5–1.99), and white low expression (<1.5).

**Figure 4 cancers-17-01958-f004:**
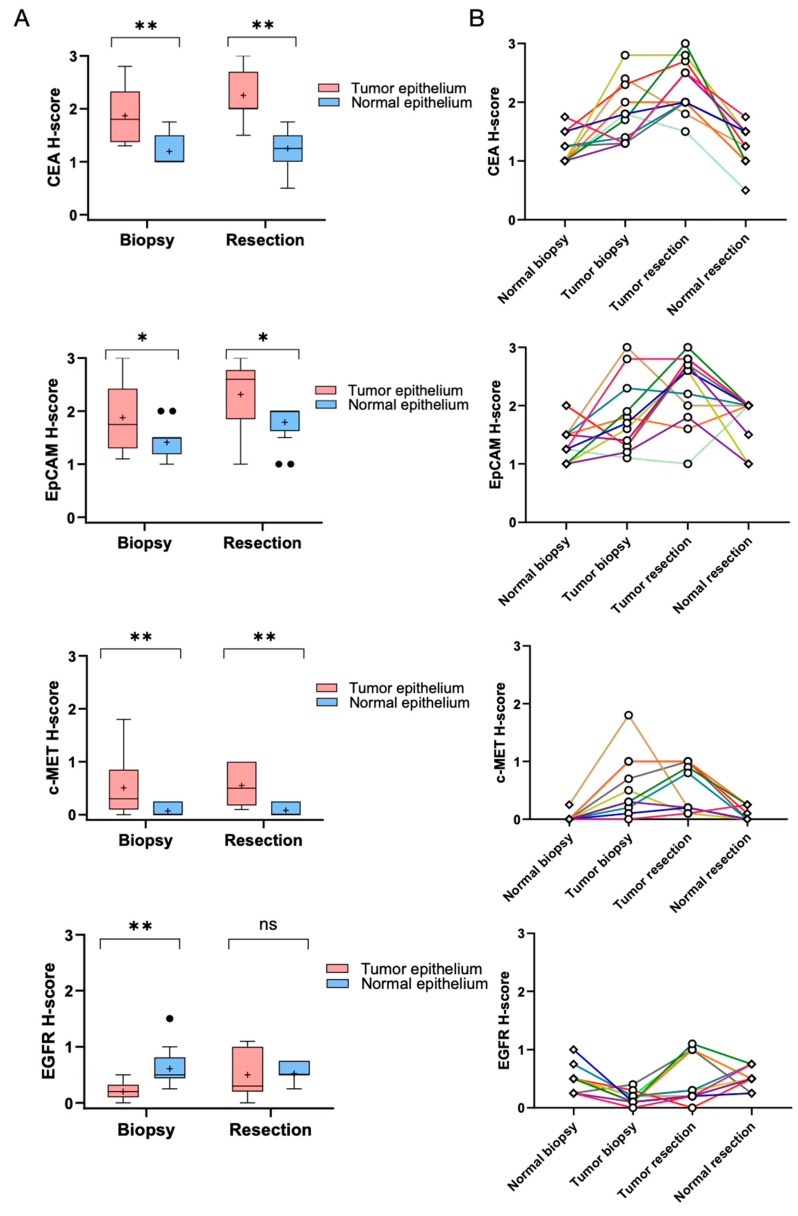
Expression changes of CEA, EpCAM, c-MET, and EGFR in paired tumor and normal tissues before and after CRT. The expression level of CEA, EpCAM, c-MET, and EGFR by comparing H-scores in paired tumor and adjacent normal epithelial tissues at two time points: diagnostic biopsies (pre-CRT) and resected specimens (post-CRT). (**A**) Box plots display the distribution of H-scores for each marker in tumor tissues compared to adjacent normal epithelial tissues. The box plots include the first quartile (Q1), median, third quartile (Q3), and maximum values. Outliers are marked with black dots (●), while the plus sign (+) represents the mean. Error bars indicate the data range, and statistically significant differences are denoted by asterisks (*). Significant differences between tumor and normal tissues were determined using the Wilcoxon signed-rank test. (**B**) Line graph shows individual changes in marker expression across patients. Each colored line represents changes in H-scores for an individual patient, showing the trajectory of expression for tumor and adjacent normal tissues before (indicated by ‘Normal biopsy’ and ‘Tumor biopsy’) and after CRT (indicated by ‘Normal resection’ and ‘Tumor resection’). A horizontal line represents equal expression levels between tumor and normal tissues, while ascending or descending lines indicate higher or lower expression in tumor tissues, respectively. Patients included in this analysis had H-scores available for both tumor and normal tissues at both time points. Abbreviations: chemoradiotherapy, CRT; ns, not significant; * *p* < 0.05; ** 0.001 ≤ *p* < 0.01.

**Figure 5 cancers-17-01958-f005:**
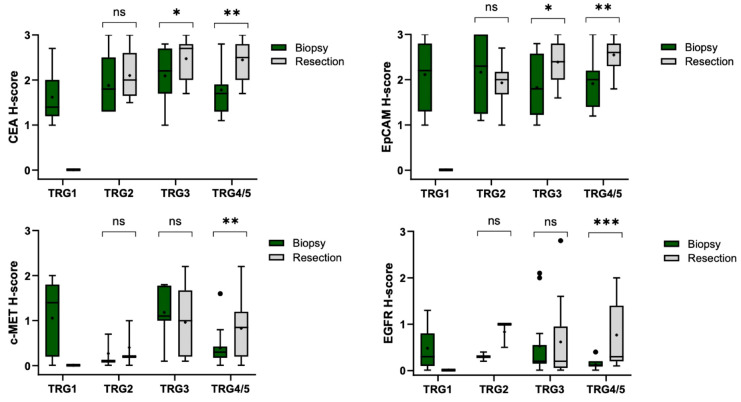
Histological score (H-score) comparison for tumor markers between pre-treatment biopsy and post-treatment specimens in various response groups to treatment. Box plots illustrate first quartile (lower part), median, third quartile (upper part), and maximum values. The outliers are shown by the black dots (●). The plus sign (+) represents the mean and error bars represent the data range. Comparison between biopsy and resection was performed using Wilcoxon signed-rank test. Statistically significant differences in expression levels are denoted by asterisks (*). For TRG1 patients, no residual tumor was present in the resection specimens. Abbreviations: TRG, tumor regression grade; TRG1, grade 1 (complete response); TRG2, grade 2 (near complete response); TRG3, grade 3 (moderate response); TRG4/5, grade4/5 (minimal to no response); ns, not significant; * *p* < 0.05; ** 0.001 ≤ *p* < 0.01; *** *p* < 0.001.

**Table 1 cancers-17-01958-t001:** Summary of the immunohistochemistry methodology. This table provides an overview of the specifications of the antibodies and methods employed in the immunohistochemistry staining for each biomarker, applied to the rectal cancer tissues.

Biomarker	Origin	Primary Antibody	Stock	Dilution	Antigen Retrieval	Secondary Antibody	Positive Control
CEA	SantaCruz Biotechnology	CI-P83–1	0.2 mg/mL	1:1000	Target retrieval solution, pH 6.0, 95 °C, 10 m (Dako PT)	Anti-mouse (Envision, Dako)	Colon tumor
EpCAM	Acris Antibodies	MOC31	0.64 mg/mL	1:10,000	Target retrieval solution, pH 6.0, 95 °C, 10 m (Dako PT)	Anti-mouse (Envision, Dako)	Colon tumor
c-MET	Bio SB	EP1454Y	1 µg/mL	1:40	Target retrieval solution, pH 9.0, 95 °C, 10 m (Dako PT)	Anti-rabbit (Envision, Dako)	Colon tumor
EGFR	Cell Signaling	D38BXP	0.64 mg/mL	1:60	Target retrieval solution, pH 9.0, 95 °C, 10 m (Dako PT)	Anti-rabbit (Envision, Dako)	Placenta

Abbreviations: CEA, carcinoembryonic antigen; EpCAM, epithelial cell adhesion molecule; c-MET, mesenchymal–epithelial transition factor; EGFR, epidermal growth factor receptor; m, minutes.

**Table 2 cancers-17-01958-t002:** Rectal cancer patient and tumor characteristics. Summary of patient demographics, tumor types, neoadjuvant therapies, surgical methods, tumor response (TRG), clinical and pathological stages, and timing between therapy and surgery.

Characteristic		Value
Patients, *n* (%)		51 (100)
Age at surgery, mean (sd), range (years)		65.55 (8.97), 40–87
Gender, *n* (%)	Male	31 (60.8)
	Female	20 (39.2)
Tumor type, *n* (%)	Adenocarcinoma	48 (94.1)
	Intramucosal carcinoma	1 (2.0)
	Mucinous	1 (2.0)
	Unknown	1 (2.0)
Type of neoadjuvant therapy *n* (%)	Capecitabine +25 × 2 Gy	26 (51.0)
	N.A. + 5 × 5 Gy	16 (31.4)
	CapOx/Avastin + 5 × 5 Gy	5 (9.8)
	CapOx+ 5 × 5 Gy	2 (3.9)
	N.A. + 13 × 3 Gy	1 (2.0)
	CapOx/Avastin + N.A.	1 (2.0)
Type of surgery	Low anterior resection	34 (66.7)
	Abdominoperineal resection	16 (31.4)
	Recto-sigmoid resection	1 (2.0)
Response *n* (%)	Complete response (TRG1)	15 (25.4)
	Near complete response (TRG2)	8 (13.6)
	Moderate response (TRG3)	13 (22)
	Minimal to no response (TRG4/5)	15 (25.4)
Clinical stage before NT	Tumor stage, *n*	
	cTx	5 (9.8)
	cT1	1 (2)
	cT2	12 (23.5)
	cT3	29 (56.9)
	cT4	4 (7.8)
	Nodal stage, *n*	
	cNx	7 (13.7)
	cN0	12 (23.5)
	cN1	21 (41.2)
	cN2	11 (21.6)
	Metastatic stage, *n*	
	cMx	36 (70.6)
	cM0	8 (15.7)
	cM1	7 (13.7)
Clinical stage after NT	Tumor stage, *n*	
	cTx	2 (3.9)
	cT0	3 (5.9)
	cT1	1 (2.0)
	cT2	5 (9.8)
	cT3	14 (27.5)
	cT4	2 (3.9)
	N.A.	24 (47.1)
	Nodal stage, *n*	
	cN0	13 (25.5)
	cN1	8 (15.7)
	cN2	4 (7.8)
	N.A.	26 (51.0)
	Metastatic stage, *n*	
	cM0	3 (5.9)
	cM1	1 (2.0)
	N.A.	47 (92.2)
Pathological stage	Tumor stage, *n*	
	pT0	16 (31.4)
	pT1	2 (3.9)
	pT2	9 (17.6)
	pT3	17 (33.3)
	pTx	1 (2.0)
	N.A.	6 (11.8)
	Nodal stage, *n*	
	pN0	31 (60.8)
	pN1	11 (21.6)
	pN2	3 (5.9)
	N.A.	6 (11.8)
	Metastatic stage, *n*	
	pM1	1 (2.0)
	N.A.	50 (98.0)
Time between neoadjuvant therapy and surgery, median weeks (range; sd)		15.1 weeks (range 1.0–91.1; 13.7)

Values are presented as number (percentage), i.e., *n* (%). Abbreviations: Gy, Gray; *n*, number of patients; c, clinical; p, pathological; NT, neoadjuvant therapy; N.A., not available; TRG, tumor regression grade.

**Table 3 cancers-17-01958-t003:** Localization of CEA, EpCAM, c-MET and EGFR expression determined by immunohistochemical staining in rectal cancer tumor tissue and healthy tissue.

Tumor Marker	Expression Location in Tumor	Expression Location in Healthy Adjacent Mucosa	Expression in Other Healthy Tissue
CEA	Apical staining	Weak to moderate staining on the apical side of 97.36% of epithelial cells, diminishing in deeper layers	No detectable expression
EpCAM	Memberanous staining	Moderate to strong membranous staining in 94.73% of epithelial cells	No detectable expression
c-MET	Memberanous and cytoplasmatic staining	Weak staining in 31.58% of luminal side of epithelial cells	No detectable expression
EGFR	Cytoplasmic staining	Weak to moderate staining in 97.37% of luminal epithelial cells; diffuse staining in deeper layers	Weak expression in 100% of muscular layers

**Table 4 cancers-17-01958-t004:** Histological score (H-score) of tumor markers in rectal cancer biopsy and surgical resection samples.

H-Score *n* (%)							
Tumor Marker	Number of Tissues (*n*)	Histology of Epithelium	No Expression (H-Score = 0)	Weak Expression (H-Score = 0.5–1.4)	Moderate Expression (H-Score = 1.5–2.4)	Strong Expression (H-Score = 2.5–3)	*p* *
CEA							
Biopsy	51	*Tumor*	0 (0%)	18 (35.3%)	23 (45.1%)	10 (19.6%)	*p* < 0.001 ↑
	19	*Normal*	0 (0%)	14 (73.7%)	5 (26.3%)	0 (0%)	
Surgical resection	31	*Tumor*	0 (0%)	0 (0%)	14 (45.2%)	17 (54.8%)	*p* < 0.001 ↑
	18	*Normal*	0 (0%)	10 (55.6%)	8 (44.4%)	0 (0%)	
EpCAM							
Biopsy	50	*Tumor*	0 (0%)	14 (28%)	19 (38%)	17 (34%)	*p* = 0.003 ↑
	19	*Normal*	0 (0%)	6 (31.6%)	13 (68.4%)	0 (0%)	
Surgical resection	34	*Tumor*	0 (0%)	1 (2.9%)	19 (55.9%)	14 (41.2%)	*p* < 0.001 ↑
	18	*Normal*	1 (5.6%)	3 (16.7%)	13 (72.2%)	1 (5.6%)	
c-MET							
Biopsy	51	*Tumor*	4 (7.8%)	35 (68.6%)	12 (23.5%)	0 (0%)	*p* < 0.001 ↑
	19	*Normal*	13(68.4%)	6 (31.6%)	0 (0%)	0 (0%)	
Surgical resection	29	*Tumor*	2 (6.9%)	23 (79.3%)	4 (13.8%)	0 (0%)	*p* = 0.005 ↑
	16	*Normal*	6 (37.5%)	10 (62.5%)	0 (0%)	0 (0%)	
EGFR							
Biopsy	51	*Tumor*	5 (9.8%)	44 (86.3%)	2 (3.9%)	0 (0%)	*p* < 0.001 ↓
	19	*Normal*	0 (0%)	18 (94.7%)	1 (5.3%)	0 (0%)	
Surgical resection	31	*Tumor*	3 (9.7%)	23 (74.2%)	4 (12.9%)	1 (3.2%)	*p* = 0.151 ↓
	18	*Normal*	0 (0%)	13 (72.2%)	5 (27.8%)	0 (0%)	

Abbreviations: TRG, tumor regression grade; CEA, carcinoembryonic antigen; EpCAM, epithelial cell adhesion molecule; c-MET, mesenchymal–epithelial transition factor; EGFR, epidermal growth factor receptor; *n*, number of patients; *p*, *p*-value. * *p*-values were calculated using the Mann–Whitney test and represent the differences in biomarker expression between tumor and normal tissue in pre-treatment biopsy and post-treatment resection samples. ↑ indicates overexpression in tumor, while ↓ represents downregulation in tumor (higher expression in normal tissue).

## Data Availability

Data supporting the findings of this study are available from the corresponding author upon request.

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
