# Peer review of "Impact of Neoadjuvant Treatment on Target Expression in Rectal Cancer for Near-Infrared Tumor Imaging"

_cancers, 2025, doi:10.3390/cancers17121958_

Round 1
Reviewer 1 Report
Comments and Suggestions for Authors
The authors address the question of using immunohistochemical biomarkers that are differentially expressed between rectal adenocarcinoma and adjacent normal tissue to help assess treatment response after total neoadjuvant therapy. They find that CEA, EpCAM, and c-MET are overexpressed in tumor tissue with c-MET and CEA showing the most promise as tumor imaging targets. While the study presents interesting findings, I have a few concerns as follows:
- The authors note that CEA, EpCAM, and c-MET are overexpressed in over 90% of tumor biopsies. However, they define a tumor-to-normal (T/N) ratio of 2 or greater as sufficient for differentiation. For CEA and EpCAM, only 26.3% and 16% of samples meet this threshold while for c-MET, it is also only 53%. How can the authors claim the over 90% figure a significant proportion of samples do not meet the T/N expression ratio considered sufficient for differentiation?
- In describing TNT, the authors do not provide adequate detail regarding radiation and chemotherapy regimen. In particular, the authors do not differentiate between radiation with concurrent chemo or sequential chemo. Short course radiation is not generally given with concurrent chemo, so it would be very surprising that 7 pts received short course RT with concurrent CapOx/Avastin or CapOx. Do the authors mean sequential chemo? For long course chemoradiation, capecitabine is generally given concurrently, but there is also a sequential chemotherapy component. The authors note that 26 pts received long course chemoradiation with capecitabine but do not identify the sequential chemotherapy component. Long course chemoRT with capecitabine by itself is not considered TNT.
- It is unclear what the significance of the authors findings is. How do the biomarkers add anything that we do not already know from the standard pathologic assessment?
Author Response
We thank the reviewer for the constructive comments and suggestions, which helped improve the clarity and clinical relevance of our manuscript.
Comment 1:
“The authors note that CEA, EpCAM, and c-MET are overexpressed in over 90% of tumor biopsies. However, they define a tumor-to-normal (T/N) ratio of 2 or greater as sufficient for differentiation. For CEA and EpCAM, only 26.3% and 16% of samples meet this threshold while for c-MET, it is also only 53%. How can the authors claim the over 90% figure a significant proportion of samples do not meet the T/N expression ratio considered sufficient for differentiation?”
Response:
We clarified this distinction in the Discussion section. Specifically, we now distinguish between general overexpression (based on positive IHC staining) and sufficient tumor-to-normal contrast (T/N ≥ 2), which is essential for imaging purposes.
Added text (4. Discussion, lines 403–407):
“While more than 90% of the tumor specimens showed positive staining for these markers, only a subset exhibited a T/N expression ratio of ≥ 2, which is considered the threshold for adequate imaging contrast. This discrepancy highlights that although overall overexpression is prevalent, only a proportion of cases exhibit sufficient differential contrast for effective fluorescence-guided surgery.”
Comment 2:
“In describing TNT, the authors do not provide adequate detail regarding radiation and chemotherapy regimen. In particular, the authors do not differentiate between radiation with concurrent chemo or sequential chemo. Short course radiation is not generally given with concurrent chemo, so it would be very surprising that 7 pts received short course RT with concurrent CapOx/Avastin or CapOx. Do the authors mean sequential chemo? For long course chemoradiation, capecitabine is generally given concurrently, but there is also a sequential chemotherapy component. The authors note that 26 pts received long course chemoradiation with capecitabine but do not identify the sequential chemotherapy component. Long course chemoRT with capecitabine by itself is not considered TNT.”
Response:
We revised the Results section to clearly differentiate concurrent and sequential administration and confirm that short-course RT was not combined with concurrent chemotherapy.
Revised paragraph (3. Results, lines 210–221):
“Patients received different forms of neoadjuvant therapy, either as part of a long-course or short-course strategy. Long-course chemoradiotherapy (LCCRT) was administered in 26 patients, consisting of capecitabine given concurrently with 25 × 2 Gy fractions of radiotherapy. Short-course radiotherapy (SCRT), defined as 5 × 5 Gy, was administered in 16 patients. In all of these cases, chemotherapy (CapOx or CapOx plus bevacizumab) was given sequentially either before or after RT as part of a TNT regimen. No patients received chemotherapy concurrently with SCRT. Additional neoadjuvant regimens included 5 patients treated with CapOx/Avastin and SCRT, 2 with CapOx and SCRT, one with a 13 × 3 Gy radiotherapy protocol, and one with CapOx/Avastin followed by other agents. Overall, 33 of 51 patients received neoadjuvant chemotherapy in combination with radiotherapy consistent with TNT.”
Comment 3:
“It is unclear what the significance of the authors findings is. How do the biomarkers add anything that we do not already know from the standard pathologic assessment?”
Response:
We added the following to the end of the Discussion section.
(4. Discussion, lines 506–512):
“While standard histopathology provides definitive insight into tumor regression after resection, it cannot inform real-time clinical decision-making prior to or during surgery. In contrast, tumor-targeted fluorescence imaging has the potential to detect residual disease in vivo during endoscopic assessment, particularly in near-complete responders. Therefore, molecular markers such as CEA and c-MET may serve as crucial adjuncts to conventional imaging by guiding preoperative response evaluation and patient selection for organ-preserving strategies.”
Reviewer 2 Report
Comments and Suggestions for Authors
This is a well-executed and methodologically rigorous study investigating the impact of neoadjuvant chemoradiotherapy (nCRT) on molecular target expression in rectal cancer tissues for the purpose of enhancing near-infrared fluorescence imaging. The rationale is clearly justified, and the study addresses a relevant clinical challenge—accurate detection of residual disease in the setting of watch-and-wait strategies. The selection of biomarkers (CEA, EpCAM, c-MET, EGFR) is appropriate, and the analysis of paired biopsy and resection samples strengthens the conclusions. The findings are supported by strong quantitative data and detailed immunohistochemical assessments, though a more explicit discussion on sample selection criteria and their potential influence on T/N ratio distributions would improve interpretability. Also, the inclusion of only 11 resection specimens in the heatmap for dual-marker analysis may underpower its generalizability and should be addressed with a short justification.
Overall, the manuscript is well-written and the data are convincing, but a few points require clarification. First, the timeline from biopsy to resection varies greatly (up to 91 weeks), and this variability might have influenced marker expression; some brief elaboration is needed on how this was handled analytically. Second, some figures (e.g., Fig. 2 and Fig. 3C) could benefit from more precise legends and improved contrast in color coding to facilitate interpretation. Lastly, minor grammar issues are present ("pose challenges" should be "poses challenges," "demonstrated safety and feasibility... conducted by our group" could be tightened for clarity), but they do not impede comprehension. The manuscript holds translational potential and merits publication after minor revisions.
Author Response
We thank Reviewer 2 for the detailed and constructive comments. All suggestions have been addressed as follows:
Comment:
“This is a well-executed and methodologically rigorous study investigating the impact of neoadjuvant chemoradiotherapy (nCRT) on molecular target expression in rectal cancer tissues for the purpose of enhancing near-infrared fluorescence imaging. The rationale is clearly justified, and the study addresses a relevant clinical challenge—accurate detection of residual disease in the setting of watch-and-wait strategies. The selection of biomarkers (CEA, EpCAM, c-MET, EGFR) is appropriate, and the analysis of paired biopsy and resection samples strengthens the conclusions. The findings are supported by strong quantitative data and detailed immunohistochemical assessments, though a more explicit discussion on sample selection criteria and their potential influence on T/N ratio distributions would improve interpretability. Also, the inclusion of only 11 resection specimens in the heatmap for dual-marker analysis may underpower its generalizability and should be addressed with a short justification.
Overall, the manuscript is well-written and the data are convincing, but a few points require clarification. First, the timeline from biopsy to resection varies greatly (up to 91 weeks), and this variability might have influenced marker expression; some brief elaboration is needed on how this was handled analytically. Second, some figures (e.g., Fig. 2 and Fig. 3C) could benefit from more precise legends and improved contrast in color coding to facilitate interpretation. Lastly, minor grammar issues are present (“pose challenges” should be “poses challenges,” “demonstrated safety and feasibility… conducted by our group” could be tightened for clarity), but they do not impede comprehension. The manuscript holds translational potential and merits publication after minor revisions.”
Responses:
- Introduction clarity and aim (1. Introduction, lines 110–117): We revised the Introduction to emphasize the impact of neoadjuvant therapy on biomarker expression and the clinical value of NIR imaging in supporting conservative strategies such as watch-and-wait.
- Sample selection and T/N ratio (2. Discussion, lines 403-414): This issue was addressed in the Discussion section. We clarified that while over 90% of tumor samples showed marker overexpression via IHC, only a subset achieved a tumor-to-normal (T/N) expression ratio ≥2, which is the imaging-relevant threshold. We also noted that this ratio could only be calculated in samples with paired tumor and adjacent normal tissue, thus influencing the representativeness of the data.
- Heatmap sample size (Figure 3 legend, lines 325–328): A statement was added clarifying that only 11 resection specimens had complete paired data suitable for dual-marker heatmap analysis.
- Biopsy-to-resection interval (2.1 Patient Cohort, lines 140–145; 4. Discussion, lines 446–448): We reported this time range (1–91 weeks) and acknowledged the potential influence on marker expression, while noting stratified analysis was not feasible due to limited samples.
- Figure legends: We improved clarity and color threshold descriptions in Figures 2 (lines 294–296) and 3 (lines 327–328).
- Grammar and clarity: Minor grammatical revisions were made throughout the manuscript, e.g., “pose” changed to “poses” (line 481).
Reviewer 3 Report
Comments and Suggestions for Authors
The manuscript presents a well-executed and timely investigation that should set the ground for a novel non-invasive approach for monitoring treatment response in rectal cancer using molecular imaging. The study’s emphasis on changes in target expression following neoadjuvant therapy, and its implications for near-infrared imaging, offer strong clinical relevance and translational promise, particularly as total neoadjuvant therapy becomes more widely adopted in clinical settings. The rationale and methodology are well-aligned with the growing emphasis on precision oncology and image-guided treatment strategies. This work provides important groundwork for integrating NIR imaging into clinical follow-up protocols for rectal cancer.
The experimental design is sound and the results are presented clearly and effectively. However, two points warrant improvement: reference to related work and clarification of sample methodology.
The discussion omits the relevant study by Galema et al. (10.1007/s11307-024-01962-6), which employs a similar conceptual framework in a different cancer type. Given the close alignment in design, this reference should be cited in the Introduction to contextualize the current study as either a complementary or derivative investigation. Including this work in the discussion would also enhance the manuscript by emphasizing the novelty of applying the approach specifically to rectal cancer and by drawing meaningful comparisons across cancer types regarding treatment-induced changes in target expression.
Section 2.1 requires further clarification. The manuscript refers to 19 tissue blocks and 59 patients but does not clearly explain the relationship between these figures. Additionally, given the long sampling period, it is important to specify how the selected samples were chosen from the larger cohort. The criteria for inclusion, whether randomized or selected based on predefined factors, should be explicitly stated to ensure transparency and reproducibility.
Author Response
We sincerely thank Reviewer 3 for recognizing the value of our study and offering valuable feedback to strengthen it further.
Comment:
“The manuscript presents a well-executed and timely investigation that should set the ground for a novel non-invasive approach for monitoring treatment response in rectal cancer using molecular imaging. The study’s emphasis on changes in target expression following neoadjuvant therapy, and its implications for near-infrared imaging, offer strong clinical relevance and translational promise, particularly as total neoadjuvant therapy becomes more widely adopted in clinical settings. The rationale and methodology are well-aligned with the growing emphasis on precision oncology and image-guided treatment strategies. This work provides important groundwork for integrating NIR imaging into clinical follow-up protocols for rectal cancer.
The experimental design is sound and the results are presented clearly and effectively. However, two points warrant improvement: reference to related work and clarification of sample methodology.
The discussion omits the relevant study by Galema et al. (10.1007/s11307-024-01962-6), which employs a similar conceptual framework in a different cancer type. Given the close alignment in design, this reference should be cited in the Introduction to contextualize the current study as either a complementary or derivative investigation. Including this work in the discussion would also enhance the manuscript by emphasizing the novelty of applying the approach specifically to rectal cancer and by drawing meaningful comparisons across cancer types regarding treatment-induced changes in target expression.
Section 2.1 requires further clarification. The manuscript refers to 19 tissue blocks and 59 patients but does not clearly explain the relationship between these figures. Additionally, given the long sampling period, it is important to specify how the selected samples were chosen from the larger cohort. The criteria for inclusion, whether randomized or selected based on predefined factors, should be explicitly stated to ensure transparency and reproducibility.”
Responses:
- Galema et al. citation:
- Introduction (lines 107–109): “Similar work in esophageal cancer by Galema et al. has demonstrated that neoadjuvant therapy can markedly influence target expression for NIR fluorescence imaging, reinforcing the need to assess these changes within each cancer type [27].”
- Discussion (lines 472–476): “Comparable translational work was recently reported by Galema et al., who investigated the impact of neoadjuvant therapy on target expression for NIR imaging in esophageal cancer. Their findings support the feasibility of imaging-based response monitoring across gastrointestinal tumors and further underscore the relevance of our study in the rectal cancer setting [27].”
- Sample methodology clarified (2.1, lines 125–131): “Tissue blocks containing both tumor and adjacent healthy tissue were selected for analysis when available; in total, 19 such blocks from a larger cohort of 59 patients were included, allowing for direct comparative evaluation. These 19 patients were selected based on the availability of paired diagnostic biopsy and resection specimens containing both tumor and adjacent normal mucosa. Selection was based on tissue quality and completeness of clinical annotation, rather than randomization.”
Reviewer 4 Report
Comments and Suggestions for Authors
Dear Author,
Thank you for sharing your research.
This study evaluated the potential of near-infrared fluorescence imaging to detect the presence of residual tumor tissue by analyzing 51 pre- and post-nCRT samples. Immunohistochemistry was used to assess the expression of four molecular markers: CEA, EpCAM, EGFR, and c-MET. CEA and c-MET were consistently overexpressed in tumor tissue, with a tumor-to-normal (T/N) expression ratio ≥2 in 45% of post-treatment samples, indicating their suitability for imaging. EGFR showed no overexpression, while EpCAM demonstrated limited utility. Neoadjuvant therapy increased marker expression in non-responders but had little impact in near-complete responders. The findings identify CEA and c-MET as promising targets for fluorescence-guided endoscopy, offering a tool for more accurate patient selection.
I found the manuscript well-written and the study particularly interesting. Currently, predicting relapse in rectal cancer remains a major challenge, especially within conservative treatment strategies. Your work offers valuable insights and could help address this important gap in knowledge.
Author Response
We thank Reviewer 4 for the thoughtful and encouraging comments.
Comment:
“Dear Author, Thank you for sharing your research. This study evaluated the potential of near-infrared fluorescence imaging to detect the presence of residual tumor tissue by analyzing 51 pre- and post-nCRT samples. Immunohistochemistry was used to assess the expression of four molecular markers: CEA, EpCAM, EGFR, and c-MET. CEA and c-MET were consistently overexpressed in tumor tissue, with a tumor-to-normal (T/N) expression ratio ≥2 in 45% of post-treatment samples, indicating their suitability for imaging. EGFR showed no overexpression, while EpCAM demonstrated limited utility. Neoadjuvant therapy increased marker expression in non-responders but had little impact in near-complete responders. The findings identify CEA and c-MET as promising targets for fluorescence-guided endoscopy, offering a tool for more accurate patient selection.
I found the manuscript well-written and the study particularly interesting. Currently, predicting relapse in rectal cancer remains a major challenge, especially within conservative treatment strategies. Your work offers valuable insights and could help address this important gap in knowledge.”
Response:
We thank the reviewer for this supportive assessment. No additional revisions were required based on this review, but we appreciate the recognition of the translational potential of our work and its relevance for guiding conservative treatment strategies.
Round 2
Reviewer 1 Report
Comments and Suggestions for Authors
The authors have adequately addressed my concerns and have improved the clarity of the findings in the manuscript.